# Five Decades of HBV Infection in Italy: A Continuous Challenge

**DOI:** 10.3390/biology12081075

**Published:** 2023-08-02

**Authors:** Tommaso Stroffolini, Giacomo Stroffolini

**Affiliations:** 1Department of Tropical and Infectious Diseases, Policlinico Umberto I, 00186 Rome, Italy; tommaso.stroffolini@hotmail.it; 2Department of Infectious-Tropical Diseases and Microbiology, IRCCS Sacro Cuore Don Calabria Hospital, Via Don A. Sempreboni, 5, Negrar, 37024 Verona, Italy

**Keywords:** HBV, vaccination, prevention

## Abstract

**Simple Summary:**

Hepatitis B virus (HBV) is a potentially life-threatening viral infection that primarily affects the liver. Vaccination against HBV is crucial for preventing the spread of the disease and reducing its associated morbidity and mortality. The introduction of the hepatitis B vaccine has had a significant impact on public health globally, including in Italy. In the 1980s, Italy had one of the highest rates of HBV infection in Europe. The Italian government recognized the importance of vaccination and introduced the hepatitis B vaccine into the national immunization program in 1991, making it freely available. This was a significant step in the fight against HBV infection. By the early 2000s, the prevalence of chronic HBV infection in Italy had decreased to less than 1% among children and adolescents. The introduction of the hepatitis B vaccine in Italy has played a crucial role in reducing the burden of HBV infection and its associated complications. The success of the vaccination program highlights the importance of public health interventions in preventing the spread of infectious diseases.

**Abstract:**

In Italy, Hepatitis B virus (HBV) infection has been characterized by several changes over the last five decades. In 2019, the incidence of acute HBV among subjects targeted by the vaccination campaign was 0 cases in the age group 0–14 years and 0.1/100,000 in the age group 15–24. Nowadays, the burden of different stages of HBV-related chronic liver diseases is minimal. Intravenous drug use is no longer a risk factor (O.R. 0.7; 95% C.I. 0.5–1.02) for acquiring acute HBV; the proportion of cases reporting this exposure fell from 29.8% to 3.3% over the last two decades. The key public health intervention has been the compulsory vaccination campaign started in 1991 for infants 3 months old and 1–2 years old (the latter group for the first 12 years of the campaign). Moreover, non-immunogenic factors and the availability of effective oral antiviral drugs have played and continue to play a prominent role. The potential availability of new oral antiviral drugs with the inherent ability to eliminate the genomic HBV reservoirs may represent a further crucial step in the elimination of the virus in people that are already infected.

## 1. Introduction

The Hepatitis B virus (HBV) is a highly contagious and potentially life-threatening virus belonging to the family Hepadnaviridae. It primarily targets the liver, leading to acute or chronic hepatitis, cirrhosis, and hepatocellular carcinoma (HCC). Understanding the virology and pathogenesis of HBV has been vital for developing effective diagnostic tools, antiviral therapies, and preventive measures to combat this persistent viral threat.

Hepatitis B virus (HBV) infection remains a significant global health concern, affecting millions of people and contributing to considerable morbidity and mortality. As of 2019, the World Health Organization (WHO) estimated that there were approximately 296 million chronic Hepatitis B surface antigen (HBsAg) carriers worldwide, with 1.5 million new infections and 820,000 deaths attributed to cirrhosis and hepatocellular carcinoma (HCC) [1]. The highest burden of HBV infection is observed in the WHO Western Pacific and Africa Regions, where 116 million and 81 million people, respectively, live with chronic infection [1].

The modern history of HBV infection began in 1965 when Baruch Blumberg made a groundbreaking discovery. He identified a surface antigen for hepatitis B (HBsAg) in the blood of an Australian Aborigine individual, initially referring to it as the “Australian antigen” [2]. Blumberg’s pioneering work led to the development of diagnostic tests and the first vaccine for HBV, ultimately earning him the Nobel Prize in 1976.

Subsequent research opened new avenues of understanding the global distribution of HBV infection, its modes of transmission, and its implications in chronic liver diseases, particularly its association with the development of hepatocellular carcinoma (HCC) [3]. A new era of research opened that subsequently led to the discovery of the defective RNA Hepatitis Delta virus (HDV) in 1977 [4], a unique virus requiring the HBsAg envelope for assembly and transmission [5].

Thanks to all these data, the hepatitis B vaccine was developed by extracting HBsAg from the blood of infected individuals. This antigen was then used to produce a recombinant vaccine that triggers the immune system to produce protective antibodies against the virus.

A new important step was taken in 1980 with the evidence from a placebo-controlled trial of the unequivocal efficacy of an inactivated hepatitis B (HB) vaccine in reducing the incidence of HBV infection by 93% in high-risk homosexual men [6]. A few years later, the immune prophylaxis with hepatitis B immunoglobulin (IBIG) plus HB vaccine was shown to be safe and highly effective in preventing HBsAg carriage resulting from perinatally transmitted infection [7], which, at that time, accounted for at least 40% of the pool of chronic HBsAg carriers in highly endemic areas [8,9].

Antiviral treatment with nucleos(t)ides analogs (NAs) became available at the beginning of the 2000s for the treatment of chronic HBV infection. The ability of these drugs to suppress viral replication has had a favorable impact on the progression of chronic HBV-related liver diseases. Moreover, the suppression of HBV replication allows the control of the spread of the virus from an infected to a susceptible subject.

First- and second-generation NAs presented a consistent risk of treatment failure. A few years later, the advent of safer and more effective third-generation NA treatments (namely, entecavir and tenofovir disoproxil fumarate) represented a great therapeutic improvement [10].

In Italy, HBV infection has been characterized by several changes over the last 50 years. The evolution of this phenomenon has been documented by numerous seroepidemiological surveys in the general population, by multicenter studies recruiting hospitalized patients, and by the specific surveillance system for acute viral hepatitis. HBV has represented a continuous challenge for clinicians in daily practice and experts in the field in Italy, leading to the publication in peer-reviewed journals of a considerable body of literature that has contributed to shaping the actual knowledge on the topic.

This review reports the major advances on the topic of HBV infection in Italy over different phases from the 1970s until the 2020s and provides insights and data from other experiences that may be relevant in the future for addressing new challenges in the fight for HBV control and elimination.

## 2. Pre-Vaccination Era (1971–1990)

At the beginning of the 1970s, a multicenter study [11] showed a high prevalence (51%) of the Australian antigen in subjects with chronic hepatitis. The marker was detected by low-sensitivity laboratory assays (i.e., immune-diffusion, complement-fixation, and counter-electrophoresis). In fact, the true prevalence could have been found to be higher by applying the current ELISA assay, which was unavailable at that time. Regardless, this study represented one of the first pieces of evidence for the link between HBV and chronic liver disease (CLD). Subsequent studies [12,13,14,15] provided the following features characterizing HBV infection in Italy:a medium level of endemicity (prevalence of HBsAg in the general population >2%)a growing prevalence of HBsAg carriers with a gradient from northern to southern areasa high proportion (>60%) of HBeAg positivity among HBsAg-positive subjectsa mode of transmission more likely occurring in the family settinga high proportion of HBsAg positivity among patients with CLDa high prevalence of Delta infection among chronic HBsAg carriers.

The availability of an effective vaccine against HBV set the stage for the start of the nationwide program of immunization for all infants in Taiwan in 1984, which was progressively extended to all elementary school children by 1990. Similar campaigns were launched in Alaska for all subjects <20 years of age in 1984, and for all infants in the Gambia in 1986. The results of these campaigns are more widely reported in a subsequent paragraph.

Thus, at the end of the 1980s, Italy was considered a country at medium levels of endemicity for HBV (HBsAg prevalence in the general population >2%). Therefore, the application of a public health policy involving a potentially similar wide vaccination campaign against HBV became a debated question in the press, the general public, and the scientific institutions of the country. With the aim of providing updated findings on the HBsAg prevalence in the youngest generations, the National Health Institute (“Istituto superiore di Sanità”) promoted a survey among subjects 3–19 years old in five Italian regions, for better tailoring and targeting the policy for a potential vaccination campaign. As many as 7405 subjects were recruited by a random sampling procedure, representing one of the largest studies ever performed in these age groups among the general population in a western country to date. The results were unexpected: the overall HBsAg prevalence was 0.6% and the prevalence of any HBV marker was 2.8% [16] (Table 1), reflecting a low exposure to HBV at any young age. This figure could be attributed to the improvement in the socioeconomic determinants of health in the country, to the decreased size of families, and to the increasing use of disposable syringes over the previous year in the country.

In fact, in this survey children whose fathers had a lower educational level (a proxy of low socioeconomic conditions) and children belonging to the largest households had a 2.3-fold and 1.7-fold higher risk of HBV infection, respectively. Alternatively, the role of contaminated injective equipment in the horizontal transmission of HBV was clearly established a few years earlier in Taiwan [17]. In Italy, up to the second half of the 1970s, the administration of injections without adequate sterilization of the medical equipment was a common practice in many families. As a consequence, the subsequent increasing use of disposable syringes has affected the intrafamily spread of HBV infection. Further evidence for the decreasing exposure to HBV was provided in 1990; in a survey among 18–26-year-old army recruits, HBsAg prevalence dropped from 3.4% in 1981 [13] to 1.6% 9 years later [18]. Similar changes were reported in Greece, where the HBsAg-carrier status among army recruits decreased from 3.4% n 1980 to 0.9% in 1986 [19]. Similarly, in Japan, HBsAg prevalence in children aged 6–15 years in 1976 was 3.8%, dropping to 1.1% in children of the same age in 1985 [20]. In these countries, the observed changes in the absence of a vaccine program were attributed to improvements in the socioeconomic determinants of health, particularly during the early years of life, with a priming effect that was considered able to affect the initial dynamics of HBV infection and have a considerable impact in its control. Still, in the absence of an effective vaccination campaign, HBV infection was spreading.

## 3. The Start of the Vaccination Era (1991–2000)

The legally enforced compulsory vaccination campaign against HBV started in Italy in June 1991. The immunization policy was stepped and innovative: other than newborns of HBsAg-positive mothers (HBsAg screening being mandatory during the third trimester of pregnancy for all pregnant women, regardless of the area of birth), 3-month-old infants and 12-year-old children (limited to the first 12 years of the campaign for the latter group) were targeted. In 1992, the campaign became fully operative. The goal was to offer protection against HBV in the timeframe of only 12 years to a large proportion of young subjects before the potential onset of risky behaviors such as unsafe sexual practices and drug use, which are more likely to start in the teenage years. The ultimate objective was not only the control of acute HBV infection but also the reduction in the number of people suffering from chronic HBsAg-carrier status and its complications, i.e., advanced chronic liver disease and hepatocellular carcinoma (HCC). The impact of this ambitious program is more widely reviewed in a subsequent paragraph.

During the 1990s, several studies were performed in the general population and CLD hospitalized patients. Surveys in the general population, all performed by a random sampling procedure for enrolment, showed an overall HBsAg prevalence below 1% [21] (Table 2). Importantly, among hospitalized patients, the rate of HBsAg positivity in chronic hepatitis cases progressively dropped from 61% in 1976–1981 to 23.2% in 2001 [14,22], and, in HCC cases, from 30.5% in 1980 to 11.5% in 1998 [23,24]. These findings confirmed the decreasing trend for HBV infection, which could not be attributed to the impact of vaccination, as this policy had started only a few years earlier targeting different population groups. In addition to improvements in socioeconomic and demographic factors, other non-immunogenic measures have played a role, namely HBsAg screening for blood transfusion (started during the 1970s), education campaigns against risky practices and behaviors, and counseling against sharing drug use equipment (both started at the end of the 1980s for controlling the spread of HIV infection, a virus that was influencing public health measures).

## 4. The Era of Antiviral Drugs and the Evaluation of the HB Vaccination Campaign (2001–2022)

As reported in the introduction, the availability at the beginning of the 2000s of NA drugs with the ability to suppress viral replication not only improved the likelihood of halting disease progression but also the control of viral spreading from infected to susceptible people. Unfortunately, these drugs are unable to eliminate HBV DNA-integrated sequences from the infected hepatocytes. Interestingly, it has been recently proven that even after prolonged treatment, a small proportion of the cccDNA reservoir is constantly replenished by continued low-level HBV replication, whereas a large proportion of the cccDNA persists over time [25]. Several new drugs that target distinct steps and pathways of the HBV life cycle have been developed and are expected to enter clinical practice in the coming years [26]. The combination of these new drugs that eliminate or functionally inactivate HBV reservoirs (cccDNA and integrated HBV DNA) with third-generation NAs is one of the proposed strategies to overcome HBV’s ability to persist as a chronic infection [27].

At the same time, a special population was identified as a key target to prevent the ongoing spreading of the virus to newborns and to evaluate the efficacy of the vaccination campaign.

At the end of the first decade (2008–2009) of the millennium, the prevention of perinatally transmitted HBV infection was evaluated in a survey involving 17,260 pregnant women [28]. Of these women, 97.7% attended prenatal screening for HBsAg. Of 138 newborns from HBsAg-positive mothers, 131 received passive and active immunization, while the remaining 7 received vaccination alone. In this survey, only 0,4% of Italian women proved to be HBsAg-positive (Table 3).

All of these women were not vaccinated because they were born before the introduction of compulsory vaccination; conversely, all women belonging to age groups targeted by the vaccination campaign tested HBsAg-negative. These findings both highlight the effectiveness of the vaccination campaign for the control of mother-to-infant HBV transmission and the successful impact of the vaccine in preventing HBsAg carriage in the age groups targeted by the campaign. A further demonstration was provided 10 years later (2010–2019) in Sicily, where the HBsAg prevalence among 6896 pregnant women was 0.2% in Italian women [29]. Moreover, one of the most convincing pieces of evidence highlighting the favorable impact of vaccination policies on HBV infection was provided in a study performed first in 1996 and again in 2010 in a small southern Italian town: no subjects below 30 years of age (targeted by the vaccination campaign) tested anti-HBc-positive, while the corresponding figure in 1996 was 6.2% [29] (Table 4).

Furthermore, in 2014, a survey focusing on a distinct population of hospitalized patients with chronic liver disease (CLD) was initiated. Notably, this survey revealed a continuous decline in the prevalence of Hepatitis B surface antigen (HBsAg) among subjects hospitalized for CLD over the years. Specifically, the prevalence of HBsAg was reported as 61% during the period 1976-81 [14], which declined to 31.3% in 1982-89 [31], further dropped to 23.2% in 2001 [22], and reached 20.2% in 2014 [22] (Figure 1).

On top of this, at the beginning of the 2010s, experiences and data from countries where comprehensive vaccination campaigns had been adopted began to be published. In Taiwan, HBsAg-carrier status in children born before and after the start of the universal HB vaccination declined from 10% in 1984 to 0.9% in 2012 [32]. Similarly, after universal HB vaccination was launched in Alaska, the incidence of acute symptomatic HBV infection in persons < 20 years of age fell from 19/100,000 cases in 1981–1982 to 0/100,00 in 1993–1994; no cases of acute HBV had occurred in children since 1992. The incidence of HCC in subjects < 20 years was 3/100,000 in 1984–1988 and zero in 1995–1999, and no cases had occurred since 1999 [33]. In Gambia, infant vaccination has prevented HBsAg carriage in early adulthood [34] (Table 5).

The HB vaccination campaign in Italy has been recently proven effective in minimizing acute HBV [35]. The fall was almost complete in people targeted by the campaign: in 2019, zero cases in the age group 0–14 years and 0.1 cases per 100,000 population (with a striking 99.4% reduction) in the age group 15–24 years were reported (Table 6).

Conversely, vaccination failure leading to the inability to develop immunity against hepatitis B is an infrequent occurrence. Extensive surveillance spanning 22 years (1993–2014) revealed that only a small fraction (0.4%) of individuals who were successfully vaccinated experienced subsequent acquisition of acute hepatitis B infection [36].

Other factors should also be considered in the falling rate of acute HBV in Italy. For example, intravenous drug use is no longer considered a risk factor (O.R. = 0.7, CI 95%, 0.5–1.02); only 3.3% of cases reported this exposure, while the corresponding figure two decades earlier was 20.8% (*p* < 0.001) [35]. Sharing a household with a chronic HBsAg carrier results in the most efficient mode of HBV transmission (OR = 10.8; 95% C.I. = 7.8–14.9). As many as 40% of acute hepatitis B cases belonging to this group were in people who did not get vaccinated, despite being aware of the carrier condition of the cohabitant. Hesitancy due to fear of vaccination and a lack of awareness of the risk of becoming infected may be reliable explanations for the failure in this setting [35]. The trend over time of some non-mutually exclusive risk factors by decade is reported in Figure 2.

## 5. Current Status and Future Scenarios

In Italy, the control of HBV infection is close to being reached for the first time in a European country. The burden of different stages of HBV-related liver diseases has strongly declined. The proportion of HBeAg positivity among HBsAg carriers is as low as 6.2% [22]. Even the main means of transmission have changed over time (Table 7).

Two main points require particular attention. Firstly, a consistent proportion of chronic HBsAg carriers are still alive in Italy. Most are >50 years old, with a long-lasting history of carrier status. They acquired their HBV infection at a younger age. Vaccination may prevent infection in susceptible hosts but not in those that are already infected. The potential availability of new drugs with the ability to eliminate the genomic HBV reservoirs may represent an important tool for HBV elimination in people already exposed to the virus. Secondly, increasing migration represents a new challenge for Italy and other European countries. Migrants are exposed to various risk factors for HBV before, during, and after migration and often come from areas where HB vaccination campaigns are defective. Indeed, they may be susceptible to HBV infection and, once infected through risk behaviors, contribute to the further spread of the infection. Currently, they account for nearly one-fifth of all acute HBV cases in Italy and are three times (95% CI = 2.5–3.6) more likely than Italian natives to acquire acute HBV [35]. Testing and the active offer of the HB vaccine to susceptible migrants may be an advisable preventive measure [37]. Moreover, migrants mostly come from areas with high HBV endemicity, such as Eastern Europe, Sub-Saharan Africa, and the Far East. For example, the rate of HBsAg positivity among 3728 migrants referring to a hospital in northern Italy in the timeframe between 2006 and 2010 was 6% [38], and this same measure increased from 14.6% in the period between 2000–2009 to 45.0% in 2010–2019 in central Italy [39], and was 9.7% among African migrants at arrival in Sicily [40]. A national survey has also shown that the proportion of people from countries other than Italy having chronic HBV infection has increased more than four-fold, from 4.5% in 2001 to 19% in 2019 [41]. For this population, appropriate antiviral treatment and adequate behavioral education to minimize further sources of exposure and onward household transmission may be required.

In 2008, the Italian Society of Infectious and Tropical Disease (SIMIT) performed a large study on the factors associated with access to antiviral treatment in 3305 chronic HBsAg carriers. It was found that immigration status was associated with a reduced likelihood of antiviral treatment [42]. Eleven years later [41], country of birth was no longer a barrier to antiviral treatment (OR 2.1; 95% CI = 0.9–4.4).

Notably, the screening of high-risk children and adults and vaccination of susceptible children, combined with treatment of chronic HB infection in migrants, are promising and cost-effective interventions, but the linkage to treatment requires more attention [38,40,43]. In particular, point-of-care screening has a high rate of acceptance, is feasible (also in undocumented migrants), and should be targeted according to provenance. Case detection of HBV infection among migrants could potentially reduce HBV incidence in migrants’ contacts and in the general population by prompting vaccination of susceptible individuals and care of eligible infected patients [38,40,43].

In view of this, it has to be considered that the linkage to care and treatment may be a challenge due to multiple barriers to accessing cultural and linguistic healthcare services along all steps of the care continuum, especially in highly mobile irregular migrants [44].

Despite the growing vulnerability and complexity of the healthcare system, access to public hospitals and availability of the current effective drugs against HBV infection in Italy remain free of charge for everyone. An additional potential primary challenge for hepatitis B in Italy in the foreseeable future is the imperative of maintaining consistently high levels of vaccination coverage throughout the population. Undeniably, the introduction of the hepatitis B vaccine in Italy has shown remarkable success in reducing the burden of HBV infection. However, as seen in other countries, there is a genuine risk that complacency could set in, leading to a decline in vaccination rates over time.

A pertinent example of this potential risk comes from France, where HB vaccination began in 1994, targeting both adolescents and infants [45]. Despite an encouraging start, vaccination uptake among teenagers in France declined to 55.4% in 2011 [46]. Similarly, concerning trends have been observed in the United States, where vaccine uptake among children and adolescents between the ages of 6 and 18 years declined during the period spanning from 1999 to 2016 [47]. These observations underscore the significance of vigilance and continuous efforts to promote vaccination awareness, uptake, and adherence to recommended immunization schedules. Vaccine hesitancy poses a significant menace to the efforts aimed at controlling hepatitis B in Italy.

Another challenge is the management of chronic HBV infection, particularly among vulnerable populations such as migrants, prisoners, and people who inject drugs. These groups are at higher risk of developing chronic infection, and access to healthcare and treatment can be limited. Furthermore, there is a need to address the stigma associated with HBV infection, which can discourage people from seeking testing and treatment. Education and awareness-raising campaigns can play an essential role in reducing the stigma and promoting testing and treatment.

## 6. Conclusions

Over the past five decades, the overall picture of HBV infection in Italy has completely changed (Table 8). The key public health intervention has been the HB vaccination campaign. The combined immunization of 3-month-old infants and 12-year-old children (limited to the first 12 years of the campaign for the latter category) has generated an early immunity cohort of youths no longer susceptible to HBV infection before they reach the age of starting risky behaviors such as drug use and unsafe sexual practices. Non-immunologic preventive measures also played an important role before the availability of the vaccine: HBsAg screening for blood and transfusion started in the 1970s, socioeconomic conditions have improved massively, and counseling against unsafe sexual practices and sharing drug use equipment started with the educational campaign for the control of the HIV epidemic. Thereafter, the combination of immunological (i.e., vaccine) and non-immunological measures, and the availability of effective antiviral drugs for subjects already infected with HBV continue to have a major impact on HBV control. Thanks to the combination of these factors Italy has shifted towards a very low endemic level. This public health story may serve other countries on the path to HBV control and elimination. Despite the proven efficacy and safety of the hepatitis B vaccine, some individuals and communities may harbor doubts or concerns about vaccination, leading to hesitancy or refusal to get vaccinated. This phenomenon has been observed not only in Italy but also in various parts of the World Health Organization (WHO) European Region.

Following the World Health Organization’s (WHO) recommendation [48], universal infant vaccination programs have been successfully introduced in 190 countries worldwide. In the European region, significant progress has been made, with 94% (50 out of 53) of the countries providing routine hepatitis B (HB) vaccination to all infants by December 2019 [49]. Notably, some countries like France and Belgium have gone beyond infant immunization, also providing HB vaccination to adolescents starting in 1994 and 1999, respectively [45]. While many countries have embraced this public health initiative, some have taken longer to implement it fully. For instance, the United Kingdom adopted the policy of HB infant immunization in August 2017, marking a late but vital step in their efforts to control the disease [50].

Moreover, the impact of universal infant vaccination on reducing HBV transmission has been remarkable. Approximately 73% (35 out of 53) of countries in the European region achieved a universal infant vaccination rate exceeding 90% on average [49]. These high vaccination rates have translated into tangible improvements in disease incidence. For instance, in Flanders, a region in Belgium, the overall rate of acute HBV cases significantly decreased from 1.7 to 0.7 between 2009 and 2017, and there were no reported cases among the age groups targeted by the immunization program [51].

Similarly successful achievements are expected in other countries in the years to come. However, at the same time, inequalities exist between high- and low-income countries with respect to vaccine availability and free access to antivirals. This represents one of the major barriers to the control of HBV infection globally.

## Figures and Tables

**Figure 1 biology-12-01075-f001:**
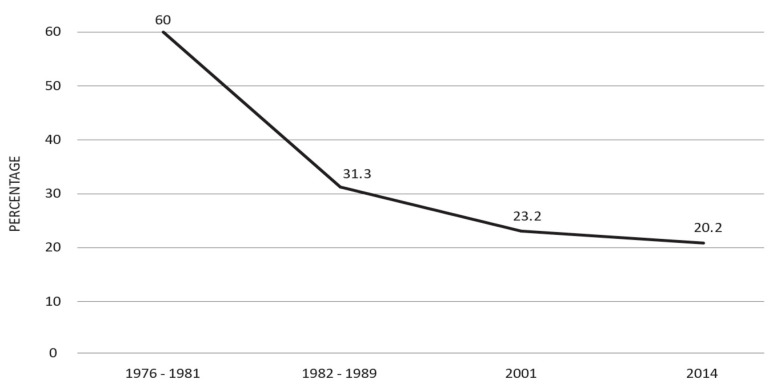
Downward trend over time in the proportion (%) of HBsAg-positive subjects among chronic liver disease (CLD) cases in Italy (adapted from References [14,22,31]).

**Figure 2 biology-12-01075-f002:**
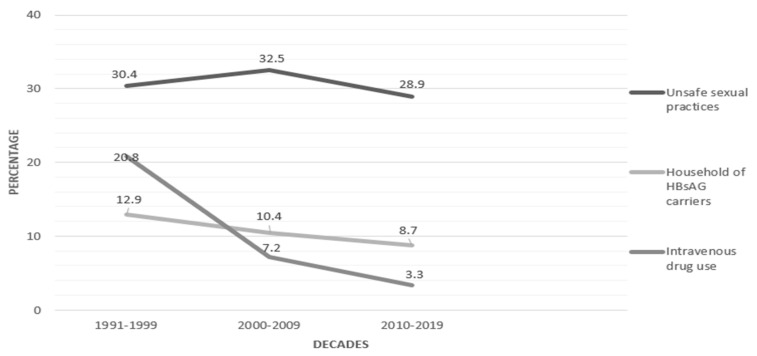
Frequency (%) of non-mutually exclusive risk factors reported for acute HBV cases by decade, 1991–2019 (adapted from Reference [35]).

**Table 1 biology-12-01075-t001:** Age-specific prevalence of HBV markers in children and teenagers in Italy 1987–1989 (Adapted from Reference [16]).

Age Group	HBsAg+ (%)	Any HBV Marker + (%)
**3–9**	0.2	1.6
**10–16**	0.7	2.9
**17–19**	0.9	4.5
**Total**	0.6	2.8

**Table 2 biology-12-01075-t002:** Prevalence (%) of HBsAg in the general population in Italian towns (Adapted from Reference [21]).

Town (Region)	Percentage (%)	Year
**Valentano (Latium)**	1	1994
**Sersale (Calabria)**	0.8	1996
**Buonalbergo (Campania)**	0.2	1997
**Camporeale (Sicily)**	0.7	1999
**Cittanova (Calabria)**	0.8	2001

**Table 3 biology-12-01075-t003:** Effectiveness of the prevention of perinatally transmitted HBV infection in Italy (adapted from Reference [28]).

	Born in Italy	Born in Africa, Asia, Eastern Europe
**n. pregnant women studied**	10,147	3229
**HBsAg prevalence**	0.4%	2.8%
**Proportion of women tested** **For HBsAg**	97.7%	97.7%
**Newborns immunized**	100%	100%

**Table 4 biology-12-01075-t004:** Age-specific prevalence (%) of antiHBc in a southern Italian town in 1996 and 2010 (Adapted from Reference [30]).

Age Group	1996	2010	*p* Value
**<30**	6.2	0	<0.01
**30–39**	17.1	3.6	<0.01
**40–49**	26.5	8.8	<0.01
**50–59**	30.3	30.3	N.S.
**>60**	32.3	32.3	N.S.
**Total**	21.5	15.2	<0.01

**Table 5 biology-12-01075-t005:** Effectiveness of HBV vaccination in areas where comprehensive campaigns have been adopted.

	Taiwan (Ref. [32])	Alaska (Ref. [33])	Gambia (Ref. [34])
**Target population**	Children < 15 y.o.	Subjects < 20 y.o.	Children
**Prevalence of HBsAg carriers**	10% (1984) to 0.9% (1999)	/	10% (1986) to 0.6% (1996)
**Acute HBV cases × 100,000**	/	10 (1982) to 0 (since 1993)	/
**HCC × 100,000**	0.52 (1984) to 0.13 (1999)	3 (1984–1988) to 0 (since 1995)	/

**Table 6 biology-12-01075-t006:** Incidence rates × 100,000 inhabitants of acute B hepatitis in Italy 1990–2019 (adapted from Reference [35]).

Age Group	1990	2019	% Reduction
**0–14**	1.0	0	−100%
**15–24**	17.0	0.1	−99.4%
**>24**	4.0	0.5	−87.5%
**Total**	5.0	0.4	−92.0%

**Table 7 biology-12-01075-t007:** The impact of non-immunogenic and immunogenic preventive measures on the modes of HBV transmission over time in Italy.

Modes of Transmission	Past	Current
**Perinatal**	+/−	-
**Child to child**	+	-
**Blood Transfusion**	++	+/−
**Intravenous drug use**	+++	+/−
**Household contact**	+++	++
**Sexual**	+++	+++

**Table 8 biology-12-01075-t008:** The changing picture of HBV infection in Italy over the past five decades. CLD: chronic liver disease. (adapted from References [21,22,31,35]).

Factors	1970s	2020s
**Endemic level**	HBsAg+ >2%	<1%
**Geographical difference**	Higher in Southern Italy	No difference
**HBeAg positivity**	High (>60%)	Low (6.2%)
**Delta infection**	Frequent	Rare
**Main mode of transmission**	Intrafamily	Sexual
**HBsAg positivity among CLD**	High	Low

## Data Availability

No new data were created or analyzed in this study. Data sharing is not applicable to this article.

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
