# Peer review of "Five Decades of HBV Infection in Italy: A Continuous Challenge"

_biology, 2023, doi:10.3390/biology12081075_

Round 1
Reviewer 1 Report
In the manuscript titled “Five decades of HBV infection in Italy: a continuous challenge”, the authors give an overview of changes in HBV epidemiology in Italy over the past decades. The manuscript reviews data that show how Italy has shifted from medium towards low HBV endemic levels.
Points to be regarded:
The introduction is too brief and without any reference. It should be expanded to contain information needed for a complete understanding of the following paragraphs, like basic information on the history of HBV infection, currently estimated prevalences of past and chronic infection in the World (from WHO publications), data on vaccine discovery and introduction and history of anti-HBV therapy. These data can then be omitted later in the text.
The manuscript contains almost 40% of self-citations and should be rewritten to include other authors’ data on HBV epidemiology in Italy.
Line 93: “HBV” should not be followed by “virus” since the “HBV” abbreviation already comprises this word.
Line 130 and throughout the text: It would be better to replace the abbreviation “HB” with “hepatitis B” since the abbreviation, despite being common, was not previously explained in the text.
The language requires minor editing.
Author Response
Dear Reviewer 1, we thank you for reviewing the manuscript and for your comments. Please find attached the point by point responses: - Introduction has been rewritten according to your suggestions. - According to your suggestions, 14 new references, not self-referred, have been added. Manuscript has been revised accordingly. - Previous line 93, word has been eliminated. - Previous line 130, the abbrevation has now been added and previously explained in the text.
Reviewer 2 Report
The review paper entitled " Five decades of HBV infection in Italy: a continuous challenge" deals with Hepatitis B prevalence reduction at a nationwide scale. There is a clear demonstration that vaccination against HBV is essential in the fightening against HBV to prevent associated liver diseases. It is known from Asian studies mainly that the introduction of HBV vaccine has a significant impact on public health globally, it is here nicely and clearly swoded for Italy. Without entering too much into details the authors may have emphasize that this model is applicable to high income countries and clearly applicable in Europe but that specific limitations exists in Southern/Low income countries that limit the vaccine implementation. On word about Vaccine Hesitancy may be provided since this phenomenon do exist especially in high income countries.
Author Response
We thank Reviewer 2 very much for his comments.
Suggestions have been added accordingly in the text
Reviewer 3 Report
The article "Five decades of HBV infection in Italy: a continuous challenge" deals with hepatitis B virus (HBV) infection in Italy.
The structure of the review article is correct and characteristic of this type of work. The authors of the text focus on presenting the changes over the years in the incidence of hepatitis B and the impact of campaigns aimed at vaccination against HBV on these changes.
The article is a fascinating work; however, as for a review article, the amount of literature reviewed by the authors could be more moderate. At the same time, out of 23 articles, the authors cited 7 of their works, in which they raise similar topics from different angles (especially citations 3, 9, 17, 21, 23). At the same time, the Pubmed database also includes publications on similar topics.
Unfortunately, despite the authors' involvement in writing the work, it should not be published in its current state and appearance because it does not bring newer information or broader commentary. A broader discussion of this topic in the context of other European countries or the impact of migration in European countries could improve the manuscript presented by the authors.
Therefore, in conclusion, I encourage the authors to edit the paper and then submit it again, because the article was read well, which proves the authors' writing skills.
Author Response
Firstly, we would like thank the reviewer for reviewing the manuscript and providing relevant suggestions; moreover, we thank the reviewer for giving us the opportunity of a second round, and in particular for having appreciated "authors writing skills". We worked on the manuscript according to reviewer suggestions. In particular: - 14 new references, not self-referred, have been added. - Discussion has been accordingly revised.Round 2
Reviewer 1 Report
The manuscript “Five decades of HBV infection in Italy: a continuous challenge” has been improved by extending the Introduction and adding new non-self references. However, The newly added references belong mostly to general knowledge in the Introduction and to very old data (from the pre-vaccination era) and are not adding any new data in the main text. Thus, all the main presented results (prevalences and their changes over the years, including all tables and figures) are still from self-cited studies. It would improve the review manuscript to include at least some data from other Italian authors on HBV prevalence in pregnant women, children, and specific patient groups, the success of anti-HBV vaccination and HBV risk factors. Though not abundant, a brief inspection of the literature shows the existence of these studies from Italy. Line 124, “HBV” should not be followed by “virus” since the “HBV” abbreviation already comprises this word. The data shown in Table 5 are missing references. The reference number 14 in the list seems not to be cited correctly.
The language requires minor editing.
Author Response
Dear reviewer, thank you for providing your comments. Ten more non self-citing references (4 from italian authors) have been added. Moreover, the cited works from italian groups cover a wide timeframe, encompassing different crucial steps about HBV knowledge in the country; references have been commented in the text. More data about special populations have been added as required. - The word "virus" has been deleted. - References have been added in the previous Table 5 (now Table 8). - Reference 14 has been amended as suggested.Reviewer 3 Report
The manuscript authors discuss the impact of hepatitis B virus (HBV) vaccination in Italy and its role in reducing the burden of HBV infection and associated complications. It highlights the historical context, mentioning that Italy had one of the highest rates of HBV infection in Europe in the 1980s. The Italian government recognized the importance of vaccination and introduced the hepatitis B vaccine into the national immunization program in 1991, leading to a significant decline in HBV infection rates.
The review mentions that by the early 2000s, the prevalence of chronic HBV infection in Italy had dropped to less than 1% among children and adolescents. It emphasizes the success of the vaccination program in reducing the incidence of acute HBV infection in different age groups. The text also notes that intravenous drug use is no longer a significant risk factor for acquiring acute HBV, and the proportion of cases reporting this exposure has significantly decreased.
The review attributes the decline in HBV infection rates to the compulsory vaccination campaign for infants and young children, which started in 1991. Additionally, it acknowledges the role of non-immunogenic factors and the availability of effective oral antiviral drugs in combating HBV infection and related liver diseases.
Overall, the review highlights the significant impact of the hepatitis B vaccination program in Italy, leading to a minimal burden of HBV-related chronic liver diseases and a decline in the risk factors associated with acquiring acute HBV infection. The potential availability of new oral antiviral drugs that can eliminate HBV reservoirs may further contribute to controlling the disease. The review underscores the importance of public health interventions, such as vaccination campaigns, in preventing the spread of infectious diseases like HBV.
However, despite the above issues, before publishing the article, I suggest a few significant changes that will improve the reception of the research.
I suggest a broader description of HBV, as well as an indication of the method of treatment or prevention in more countries. A comparison of data from decades in other countries could further develop this article.
Another issue, maybe the authors would have thought about adding additional figures showing specific data and age groups.
After making the above changes, I believe you will be able to publish the article.
Author Response
Dear reviewer,
thank you for providing your comments.
- A broader description of HBV has been provided
- Data from other European countries, and comparisons with those, have been provided and commented in the text.
- 3 new Tables and 1 Figure have been added to the text.
Round 3
Reviewer 3 Report
I would like to thank the authors for taking into account the suggestions, the article can be accepted in its current form.